# Sustainable and Clean Utilization of Yellow Phosphorus Slag (YPS): Activation and Preparation of Granular Rice Fertilizer

**DOI:** 10.3390/ma14082080

**Published:** 2021-04-20

**Authors:** Cuihong Hou, Luyi Li, Lishuang Hou, Bingbing Liu, Shouyu Gu, Yuan Yao, Haobin Wang

**Affiliations:** School of Chemical Engineering, Zhengzhou University, Zhengzhou 450001, China; hch92@zzu.edu.cn (C.H.); lly15993001690@gs.zzu.edu.cn (L.L.); hls@gs.zzu.edu.cn (L.H.); sygu@zzu.edu.cn (S.G.); yaozhikang@live.cn (Y.Y.); hb-wang@zzu.edu.cn (H.W.)

**Keywords:** yellow phosphorus slag, activation, slag system optimization, granular rice fertilizer, nutrient conversion

## Abstract

Yellow phosphorus slag (YPS) is a typical industrial solid waste, while it contains abundant silicon micronutrient required for the growth of rice. The key scientific problem to use the YPS as rice fertilizer is how to activate the slag efficiently during the phosphorite reduction smelting process. In this work, an alkaline rice fertilizer from the activated YPS was successfully prepared to use the micronutrients. Thermodynamic analyses of SiO_2_-CaO, SiO_2_-CaO-Al_2_O_3,_ and SiO_2_-CaO-Al_2_O_3_-MgO systems were discussed to optimize the acidity for reduction smelting. Results showed that the reduction smelting followed by the water quenching process can realize the reduction of phosphorite and activation of YPS synchronously. Ternary acidity m(SiO_2_)/(m(CaO) + m(MgO)) of 0.92 is suitable for the reduction smelting and activation of the slag. After smelting, the molten YPS can be effectively activated by water quenching, and 78.28% P, 90.03% Ca, and 77.12% Si in the YPS are activated, which can be readily absorbed by the rice roots. Finally, high-strength granular rice fertilizers with a particle size of Φ2–4 mm were successfully prepared from the powdery nitrogen-phosphorus-potassium (NPK) and activated YPS mixture.

## 1. Introduction

Rice is the second-most widely grown crop in the world and is also the major staple food for more than half of the world’s population. China is the biggest rice producer with a total rice output of nearly 200 million tons, accounting for 37% of the country’s total grain output and 35% of the world’s total rice output [1]. Rice is a typical Si hyper-accumulating plant species [2]. As one of the most basic nutrient elements for rice growth, silicon can promote rice growth and development; enhance its defense ability against diseases and insect pests; increase its tolerance to drought, salt stress, and heavy metals; and improve rice yield [3,4,5,6]. Generally, it is estimated that 230–470 Kg·ha^−1^ of Si was consumed when 5000 Kg·ha^−1^ of rice was produced. However, 75–80% of the absorbed silicon comes from the soil [7]. Resultantly, a long-term lack of Si in the soil will reduce rice production [2,8]. Depending on the statistics, 50% of the Chinese cultivated land, especially in the southern paddy fields with acidic soil, lacks silicon nutrient [9]. Hence, the application of silicon fertilizer is essential for the sustainable development of rice crop [10,11].

The industrial silicon fertilizer is mainly manufactured by the high-temperature smelting process. The high-purity quartz sand, cosolvent, and additives are smelted and then cooled to generate the silicon fertilizer, and the effective silicon in this fertilizer is about 50% [12]. Currently, a considerable amount of silicon slag is produced in China every year [13,14,15], and this slag faces a dual concern with respect to waste of resources and environmental pollution. Some attempts were made to utilize the industrial silicon-containing slag as the raw material to produce silicon fertilizer. However, the effective silicon content is only in the range of 15%−20% [16,17]. Some studies demonstrated that rice fertilizer prepared from silicon-containing slag is characterized as citric-soluble, alkalescent, and slow-release, which can promote the growth and disease resistance of the rice plants, especially for the acid soil [18,19,20]. Several attempts were made to utilize the solid waste and sludge to prepare crop fertilizer [21,22]. However, how to improve the activities of the silicon-containing slag directly during the high-temperature reduction smelting process so that they are economical and energy-efficient to make agricultural silicon fertilizer remains unclear.

Generally, reduction smelting of phosphate ore for yellow phosphorus production is the main treatment for phosphorus extraction. YPS, composed of 40–60% CaO, 25–42% SiO_2_, 2.5–5% Al_2_O_3_, 0.2–2.5% Fe_2_O_3_, 0.5–3% MgO, and 1–4% P_2_O_5_ [23], is a solid waste generated in the yellow phosphorus production in an electrical furnace at the temperature range of 1400–1600 °C from phosphate ore, silica, and coke [14]. In industrial production, this process consumes about 12.7 t of solid raw materials, which usually include 9.5 t of phosphate rock, 1.5 t of coke, and 1.7 t of silica to produce 1 t of yellow phosphorous [24,25], concomitantly generating 8–10 t of YPS. Some attempts were tried to use the slag to produce building materials, glass ceramics, and ceramic materials [13,26,27,28]. However, most of this slag is stockpiled in factories because of its limited utilization, currently. On the other hand, YPS contains a variety of nutrient elements needed for crop growth. Phosphorus is one of the necessary elements for crop growth. Magnesium is the main component of crop chlorophyll, which can promote photosynthesis. Calcium plays an irreplaceable role in neutralizing the acidity of the soil and can eliminate the poisoning of other ions to crops (such as heavy metals) [7]. The application of yellow phosphorous slag for high-value-added rice fertilizer production is an effective way to utilize various nutrients needed for rice growth in the slag comprehensively.

However, the key scientific problem to make YPS as rice fertilizer is how to activate the YPS effectively. Takahashi [29] showed that the availability of silicon in slag increased with the decrease in particle size, which was more conducive to the release of effective elements. Tasong et al. [30] investigated the effect of treating slag with lime on the leaching of helpful elements and obtained a better activation effect. However, this method takes time and consumes higher energy. On the other hand, some efforts are also tried to extract the phosphorus from P-bearing steel slag and high-P oolitic hematite [31,32]. Liu et al. [32] reported that P can be made for crop fertilizer, and Fe can be utilized for ferrite ceramic production from the refractory high-P oolitic hematite via high-temperature phase reconstruction. These studies demonstrate that phosphorus extraction and activation are obviously affected by temperature. In consequence, it is beneficial from economic and environmental perspectives if nutrient elements in the slag are synchronously activated during the yellow phosphorus production by adjusting the yellow phosphorus feeding formula, smelting temperature, and cooling regimes in this study.

In order to make the YPS as a special rice fertilizer to utilize the micronutrients, this study aims to improve the activities of silicon and calcium in YPS via adjusting the acidity, reaction time, smelting temperature, and the cooling regimes during the reduction smelting processing of phosphorite. Firstly, thermodynamic analyses of SiO_2_-CaO, SiO_2_-CaO-Al_2_O_3,_ and SiO_2_-CaO-Al_2_O_3_-MgO systems for YPS formation were discussed to optimize the acidity for smooth reduction smelting. Then, the phase transformation, microstructure characteristics, and the activities of nutrients were investigated. After optimization, the activated YPS was used as an ingredient to prepare the special rice fertilizer. The as-prepared granular rice fertilizer with an average particle size of 2–4 mm showed good fertilizer efficiency and mechanical strength. In addition, the toxicity characteristic leaching test of the heavy metals in the YPS was also characterized to evaluate the environmental and alimentary risks.

## 2. Experimental

### 2.1. Raw Materials

Phosphate ore, silica, and coke used in the experiment are taken from Yunnan, China. The chemical compositions of the phosphate ore and silica flux are listed in Table 1. The quantitative element compositional analysis is carried out by chemical titration. All the titration tests for each element analysis were conducted thrice, and the average value was set as the final result. The main chemical compositions of phosphate ore are 33.72% CaO, 29.04% SiO_2_, 22.57% P_2_O_5_, 3.63% Al_2_O_3_, and 1.79% MgO. XRD patterns of the phosphate ore shown in Figure 1 reveal that phosphate ore mainly consists of fluorapatite and quartz. Silica flux contains 87.30% SiO_2_, which is regarded as a cosolvent to adjust the acidity (mass ratio of SiO_2_/ (CaO + MgO) of the smelting burden and reduce the melting point of slag. Coke is acted as a reductant for the reduction of phosphorite ore to produce yellow phosphorus in the electric furnace. The fixed carbon and ash in coke account for 62.48 wt % and 28.96 wt %, respectively. As shown in Table 2, the main chemical compositions of coke ash are determined with reference to the test method for analysis of coal ash (GB/T 1574–2007) since the coke ash has a great influence on the acidity of the burden.

### 2.2. Preparation of YPS

Before reductive smelting experiments, the raw materials consisting of phosphate ore, silica, and coke were dried at 105 °C for 5 h, and then ground to 100%, passing 0.125 mm standard sieve. The chemically pure reagent CaCO_3_ (99.9 wt %) and silica are utilized to adjust the acidity, and the burdening schemes are listed in Table 3. After burdening, the raw materials are evenly mixed in an agate mortar. Then, the mixtures are put into a cylindrical heat-resistant graphite crucible with a diameter of 80 mm and height of 80 mm. The graphite crucible carrying the samples is loaded into the shaft furnace (GWL-1700 VSF-SR, Guoju Precision Electric Furnace Co., Ltd., Luoyang, Henan, China) for reduction smelting. The schematic diagram of the reduction smelting furnace is illustrated in Figure 2. The reduction smelting furnace integrating the controlling system is controlled by a microcomputer program for automatic heating, which effectively guarantees the accuracy and stability of temperature. Constant temperature accuracy and temperature control accuracy are within the range of ±3 °C. The detailed procedures of reduction smelting were described in our previous works [33,34]. After being isothermally smelted in pure N_2_ (99.99%) atmosphere at given temperatures and periods, volatile yellow phosphorus is recovered and absorbed via a water tank from the exhaust gas, while the slag residue is controllably cooled to activate the nutrients. Three kinds of cooling regimes for the slag are designed, such as water quenching, air cooling, and furnace cooling. The cooled slags were then dried at 105 °C for 4 h and ground into pulverous samples for further characterization and analysis.

### 2.3. Preparation of Rice Fertilizer Using YPS

During the plantation of rice crop, nitrogen, phosphorus, potassium, and silicon were the most basic nutrient elements to maintain the healthy growth of rice. In China, the production of yellow phosphorus is mainly concentrated in the Yunnan and Guizhou provinces. For large-scale utilization of YPS in these regions, rice fertilizer should be prepared based on the soil nutrient status and planting habits in the Yunnan and Guizhou provinces. In this study, the N-P_2_O_5_-K_2_O formula in the fertilizer is designed to be 18-8-10 and 20-8-12 based on the nutrient requirements of rice crops, which were reported by the local agronomists [35,36]. Notably, the rice fertilizer formula of 18-8-10 (N-P_2_O_5_-K_2_O) refers to that 100 Kg of fertilizer containing 18 kg of N, 8 kg of P_2_O_5_, and 10 kg of K_2_O.

Compatibility and synergistic effects of various nutrient forms are important for the fertilizer. Nitrogen in the designed rice fertilizer is provided by urea (N = 46%, Henan Xinlianxin Co. Ltd., Henan, China), monoammonium phosphate (N = 11%, P_2_O_5_ = 60%, Hubei Yishizhuang Agricultural Science and Technology Co. Ltd., Hubei, China), and ammonium sulfate (N = 20.5%, Hubei Yi Shi Zhuang Agricultural Science and Technology Co. Ltd., Hubei, China). Phosphorus in the fertilizer is offered by the monoammonium phosphate and the prepared YPS (P_2_O_5_ = 3.21%, SiO_2_ = 41.58%, and CaO = 43.34%) from reduction smelting of phosphate ore. Potassium is served by industrial-grade red potassium chloride (K_2_O = 60%, Qinghai Salt Lake Industry Co. Ltd., Qinghai, China), and silicon and calcium are also provided by the YPS. The amounts of raw materials required for the preparation of fertilizers (Sample A1 and Sample A2) to produce 1000 g rice fertilizer are listed in Table 4. In particular, the additions of activated YPS are 25% in both Samples A1 and A2.

Figure 3 shows the preparation process of rice fertilizer. Rice fertilizer is made by powder granulation technology through several steps. The first one is to make the pretreated raw material through a 120-mesh sieve, then be weighed according to the composition shown in Table 4 and evenly blended with the mixture. The next step is to put the mixed powder into the granulator to make pellets. When most of the pellets are grown with a granule size of 2–4 mm, qualified granules are screened out, and the materials less than 2 mm are returned to be re-granulated. The qualified granules are dried at a low temperature of 60 °C for 5 h and then sealed for further analysis.

### 2.4. Determination of the Activities of Nutrients

In order to simulate the absorption effect of micronutrients by crop roots, the effective content of nutrients is measured according to the Chinese standard (NYT 3034-2016) [37], Soil Conditioner. The effective SiO_2_ and CaO were determined by 0.5 mol/L HCl solution as the extractant, and then the effective SiO_2_ was measured by the potassium fluosilicate volumetric method, and the effective CaO was determined by the ethylenediaminetetraacetic acid disodium volumetric method. The sample preparation method for measuring the effective content of P_2_O_5_ and K_2_O is similar to that of the effective contents of SiO_2_ and CaO. However, there are also differences; 0.5 mol/L HCl solution was replaced by 20 g/L citric acid solution as the extractant, and the effective K_2_O and P_2_O_5_ were determined by the sodium tetraphenylborate gravimetric method and the gravimetric method of quinoline phosphomolybdate, respectively. Besides, nitrogen is analyzed by the titration method after the distillation process with K1100 automatic Kjeldahl azotometer [38]. Each experimental datum was taken from the average value of three parallel determinations. Meanwhile, the total element content was calculated through material balance. Activation efficiency (or effective conversion) of the nutrients was calculated using Equation (1).
(1)$$\eta =\frac{C}{T}\times 100\%$$
where η is the activation efficiency (or effective conversion) of the nutrients, %; *C* and *T* are the content of effective nutrient and total nutrient, respectively, %.

### 2.5. Instrumental Characterization

Phosphate ore and YPS were pre-ground to a granule size passing through a 200-mesh standard sieve before testing. X-ray diffraction (XRD) patterns of the smelted samples were detected by a diffractometer (X’ Pert Pro MPD) under the conditions of radiation: Cu Ka, scanning range: 5°–90° (2θ), step size: 0.02° and scanning speed: 5°/min. The voltage and current of the machine were set at 40 kV and 40 mA, respectively. Polished sections of the roasted samples were prepared to observe the microstructure and analyze the elemental content of the phases by using a scanning electron microscope (SEM, TESCAN, MIRA3, LMH/LMU, Czech Republic) equipped with an energy-dispersive X-ray spectroscopy (EDS) detector. SEM images were recorded in backscatter electron mode operating in a low vacuum of 0.5 Torr and 20 keV. In addition, all the external standards and matrix correction procedures for EDS are based on international standards, and the results are analyzed automatically by the instruments and software. Granule strength of special compound fertilizer for rice was determined by granule strength meter (DL3, Penghui Technology Development Co. LTD, Dalin, China). Thermodynamic calculations for YPS formation based on the method of minimization of the Gibbs’ free energy were performed at atmospheric pressure by the Factsage 7.3 software. The “Phase diagram” and “Reaction” modules were adopted, and the used databases were “FToxide” and “FTPureSubstance”.

## 3. Results and Discussion

### 3.1. Thermodynamic Analysis of YPS Formation

YPS contains a large amount of CaO, SiO_2_, Al_2_O_3,_ and MgO, and these oxides usually account for more than 90% of the total content. Therefore, the binary SiO_2_-CaO system and ternary SiO_2_-CaO-Al_2_O_3_ system are regarded as the benchmark systems for the smelting and slagging of phosphate ore. Sometimes, phosphate ore also involves 1–3%MgO, and the quaternary SiO_2_-CaO-Al_2_O_3_-MgO system should also be taken into consideration. The thermodynamic equilibrium of these systems in the temperature range of 1000–1800 °C under 90 vol % CO-10 vol % CO_2_ atmosphere is calculated, and the phase diagrams are shown in Figure 4.

As can be viewed in Figure 4a, the CaO-SiO_2_ system shows that the melting points of calcium silicate compounds formed by the reaction of CaO and SiO_2_ at different temperatures are relatively high. When the binary acidity (m(SiO_2_)/m(CaO)) ranges from 0.75 to 1.2, the main phases are tridymite (SiO_2_(S4)), rankinite (Ca_3_Si_2_O_7_), wollastonite (CaSiO_3_), and cyclowollastonite (CaSiO_3_(S2)) within the temperature scope of 1000–1400 °C. As the temperature is increased to 1460 °C, the phases are converted to the multiphase mixture including liquid, Ca_3_Si_2_O_7,_ and CaSiO_3_(S2). However, binary acidity for industrial yellow phosphorus production is usually controlled between 0.75 and 0.85 for smooth liquid slag discharge. Consequently, it is notable that the binary system with an acidity of 0.785 has a high melting temperature of 1464 °C. If the acidity is less than 0.785, a higher temperature is required to generate enough liquid phase due to the formation of Ca_2_SiO_4_ with a higher melting point.

Figure 4b,c display that the main phase of the SiO_2_-CaO-Al_2_O_3_ system with less than 5% of Al_2_O_3_ content is also Ca_2_SiO_4_ as the binary acidity increased from 0.75 to 0.95. However, the primary phase of the SiO_2_-CaO-Al_2_O_3_-MgO system changes to the Ca_3_Si_2_O_7_ with the ternary acidity (m(SiO_2_)/(m(CaO) + m(MgO))) range of 0.75–0.95. Remarkably, the lowest melting point of the SiO_2_-CaO-Al_2_O_3_-MgO system reduces about 20 °C in contrast to the ternary system, indicating that the formation of Ca_3_Si_2_O_7_ induces the low melting point. However, it is also found that the main phase is transformed to the region of Ca_3_Si_2_O_7_ and CaSiO_3_(S2) with a lower melting point because of the proper increase of acidity. These results indicate that an appropriate increase in acidity or MgO addition can facilitate the formation of the melting phase (mainly rankinite and cyclowollastonite). As a result, a higher ternary acidity (m(SiO_2_)/(m(CaO) + m(MgO))) of 0.90–0.95 is used for the reduction smelting of phosphate ore. It is economical to cut down the energy consumption of yellow phosphorus production due to the lower reduction smelting temperature. In the subsequent experiment, ternary acidity (m(SiO_2_)/(m(CaO) + m(MgO))) of 0.92 is designed.

### 3.2. Synchronous Reduction of Phosphorite and Activation of YPS

Figure 5a,b illustrates the XRD patterns of phosphorite with ternary acidity of 0.92 smelted at the temperature range of 1200–1500 °C for 60 min under different cooling regimes. It can be observed that there is no significant difference in the main phase of slag systems under different cooling regimes below 1350 °C. When the phosphorite is smelted at 1200 °C, the principal phases are Ca_5_(PO_4_)_3_F, SiO_2_ and wollastonite (CaSiO_3_). Unlike the water quenching, a small amount of the Ca_2_Al_2_SiO_7_ phase is found, as the smelted sample is cooled in the air atmosphere. As the temperature increased to 1300–1500 °C, some wollastonite (CaSiO_3_) transformed to cyclowollastonite (CaSiO_3_(S2)), and this result is consistent with the phase diagram of the SiO_2_-CaO system shown in Figure 4**.** Moreover, with the rise in temperature, the intensity of the diffraction peaks of CaSiO_3_(S2) first increases noticeably and then decreases, while that of the Ca_5_(PO_4_)_3_F and SiO_2_ declines, indicating that CaO facilitates the Ca_5_(PO_4_)_3_F reacting with SiO_2_ to form silicates.

As the phosphorite is smelted over 1400 °C, the samples are presented as an amorphous state. These XRD patterns of the samples smelted at 1450 °C are summarized in Figure 5c. Under the air cooling and water quenching, the XRD patterns of the slag have no sharp diffraction peaks, basically showing wide-range and slowly-changing diffraction peaks. It is demonstrated that YPS obtained under these conditions shows glassy substances with low chemical stability [39,40]. The glass structure of YPS after water quenching is illustrated in Figure 5d. However, as the smelted samples were cooled in the thermal furnace with a slow cooling rate, the main phases are well-crystallized CaSiO_3_, Ca(Mg_0.85_Al_0.15_)(Si_1.70_Al_0.30_)O_6_, and Ca_3_(SiO_3_)_3_ since the molten liquid has enough time to cool and crystallize. The reduction smelting followed by the water quenching process can realize the reduction of phosphorite and activation of YPS synchronously. Finally, it is suggested that highly active amorphous YPS can be obtained when the phosphorite is smelted at 1450–1500 °C for 60 min with a ternary acidity of 0.92 followed by water quenching or air cooling, and the activated slag can be used as an ingredient for the preparation of rice fertilizer.

### 3.3. Microstructure of the YPS

The microstructure of the YPS obtained at 1450 °C by water quenching and air cooling is displayed in Figure 6. It can be seen that the structure of YPS is porous owing to the phosphorus escaping during the reduction process. In particular, water quenching slag is with a thin-walled structure, which is ascribed to the drastic volume shrinkage and rupture of the high-temperature molten slag during the ultrafast cooling environment [38]. However, air cooling slag shows a relatively dense structure. SEM mapping results indicate that all nutrient elements are evenly distributed in the water-quenched slag. In general, after ultrafast water quenching, the molten slag is transformed into microporous, multi-soluble matter and an amorphous structure [41,42], which is beneficial to the excitation of YPS to release the micronutrients.

### 3.4. Activities of Nutrients in the YPS

In order to determine the activities of nutrients in the YPS, the influence of different reaction conditions such as acidity, temperature, time, and cooling regimes on the activation of silicon and calcium in YPS are studied experimentally, and the corresponding results are drawn in Figure 7. It is clearly seen in Figure 7a that the overall activation efficiency of calcium silicate, first increasing slightly and then gradually decreasing with acidity increasing from 0.6 to 1.2, presents a “peak” at an acidity of 0.92. However, a maximum activation efficiency of 93.15% for calcium and a minimum activation efficiency of 3.66% for silicon is obtained under acidity of 1.2. It is inferred that the Ca_5_(PO_4_)_3_F reacts with SiO_2_ immediately to form CaSiO_3_ resulted in the presence of excessive silica in the system. By comprehensive consideration, the acidity of 0.92 is recommended since the conventional acidity for the reduction smelting of phosphorite is in the scope of 0.75–0.85. The corresponding silica and calcium activation efficiencies are 77.12% and 90.03%, respectively. Figure 7b shows the effect of cooling regimes on the activation efficiencies of silica and calcium. In contrast to the air cooling slag and furnace cooling slag, the silicon activation efficiency of the water quenching slag is enhanced by 14.45% and 11.72%, while the activation efficiency of calcium is increased by 12.62% and 10.35%. This is further demonstrated that water quenching slag presents high activity of silicon and calcium due to the special amorphous, porous, and thin-walled structure.

Furthermore, activation efficiencies of silica and calcium in the YPS are shown in Figure 7c,d. It is found that the activation efficiencies are firstly increased dramatically and then slightly declined by prolonging the time from 20 min to 80 min and the increase of temperatures from 1200 °C to 1600 °C. It is evident that the activation efficiencies of silicon and calcium are only 62.45% and 77.30% since the reduction smelting of the burden is not complete in such a short time of 20 min, but higher temperature or longer time can give rise to a decrease in nutrient activity in slag. Appropriate increases in smelting temperature and time can improve the activation efficiency.

### 3.5. Preparation and Properties of Special Compound Fertilizer for Rice

#### 3.5.1. Toxicity Test of the Heavy Metals in the Activated YPS

Although YPS is an excellent raw material for rice fertilizer, it usually contains trace heavy metals such as chromium (Cr), cadmium (Cd), lead (Pb), mercury (Hg), arsenic (As), and so on [20,43], which are harmful to the growth of crops. Hence, it is exceedingly necessary to carry out a detailed assessment of heavy metal information about the activated YPS before fertilization. The standard of heavy metal limitation is established by the Chinese government (Chinese standard, GB 38400–2019, limitation requirements of toxic and harmful substance in fertilizers) [44], and the total concentration of heavy metals in fertilizers is usually used as the evaluation index. The comparison between the content of heavy metals in YPS and the fertilizer limitation is listed in Figure 8. As shown in Figure 8, the As, Cd, Pb, Cr, and Hg contents in the slag are 1.52 mg/kg, 0.75 mg/kg, 20.81 mg/kg, 71.10 mg/kg, and 1.02 mg/kg, respectively, and these heavy metal contents in the slag are much less than the required fertilizer limitation. Therefore, the activated YPS generated from the phosphorite ore can be used as raw material for the preparation of rice fertilizer.

#### 3.5.2. Activities of Various Nutrients in the Rice Fertilizer

Some studies showed that the silicate powder is also fed to enrich the soil together with NPK (nitrogen, phosphorus, and potassium) fertilizer, and the presence of silicon also can promote the absorption of NPK by the rice roots [18,19]. However, farmers usually blindly apply high-proportioned NPK fertilizer in pursuit of high yield in the current rice production process. Inversely, this overfertilization results in an imbalance of N, P, and K in the soil and further lowers the rice yield [45,46]. Suitable formulation of NPK is vital to rice planting. In this work, NPK fertilizers for regional rice planting with formulas of 18-8-10 (N-P_2_O_5_-K_2_O) and 20-8-12 (N-P_2_O_5_-K_2_O) are prepared according to the soil and rice planting requirements in the Guizhou and Yunnan provinces in China. Two kinds of granular rice fertilizers (sample A1 and A2) were then prepared based on the method described in Figure 3. Some qualified fertilizer granules are ground to analyze the nutrient content, and the others are used to determine the average crush strength of the granules.

Figure 9a shows that the effective contents and total contents of N, P_2_O_5_, and K_2_O in samples A1 and A2 are basically the same, and are in line with the formula ratio of 18-8-10 and 20-8-12. During the absorption simulation of the rice plant roots system (as described in Figure 9b), the N and K_2_O, both in the sample A1 and A2, have higher conversion efficiencies above 98.33%. However, the effective conversions of P_2_O_5_ in the sample A1 and A2 are 94.39% and 94.51%, which are lower than that of the N and K_2_O. This is because not all the phosphorus in the slag is effective. Uncharacteristically, unlike the conventional NPK fertilizer, this special NPK fertilizer is prepared from the activated YPS, which contains some CaO and SiO_2_ micronutrients. The effective conversions of silicon and calcium in the fertilizer are as high as about 80% and 90%, which can be readily absorbed by the rice roots. Therefore, it is economically reasonable that YPS is a satisfactory ingredient for rice fertilizer production.

#### 3.5.3. Granule Strength of the Fertilizer

High-quality fertilizer granules are required to have good mechanical strength to resist normal transportation, baggage storage, handling, and mechanical spreading without breakage [47,48]. Thirty rice fertilizer granules with a size of 2–4 mm were randomly selected to measure the compressive strength through the strength meter, and the average value of the compressive strengths was regarded as the final compressive strength. The average compressive strengths of the granular rice fertilizers are shown in Figure 10a. In general, the average crush strengths of samples A1 and A2 decrease slowly at first and then tend to be stable with the continuous extension of time. The strengths of samples A1 and A2 are finally stable at about 25 N and 21 N. As presented in Figure 10b, granular rice fertilizers show good sphericity. However, the strengths of rice fertilizer particles decrease in the first three days, which is probably due to that the moisture in the air entering the interior of the granules. The moisture inside and outside of the particles is in dynamic equilibrium, and thus the strength tends to be stabilized. Notably, the average compressive strength of sample A1 is higher than that of sample A2 since the addition of attapulgite enhances viscous forces in the liquid bridge between primary particles [49]. It is concluded that the average crush strengths of samples A1 and A2 meet the actual storage and transportation requirements of rice fertilizer products, and the appropriate addition of attapulgite during the granulation process can significantly increase the granule strength of the fertilizer.

### 3.6. Total Flowsheet and Element Distributions

Figure 11a shows the total flowsheet and nutrients distribution in the treatment of phosphorite. As presented in Figure 11, after burdening, mixing, and reduction smelting, the phosphorus is divided into three parts in which 96–97% P is in the vapor, 1–2% P is in the ferrophosphorus alloy, and 2–3% P is in the molten slag. The reduction of Ca_5_(PO_4_)_3_F to P_4_ is based on the Equation (2). The formation of ferrophosphorus alloy (FeP_2_) is derived from the stepwise reduction in Fe_2_O_3_ to metallic Fe and the alloying of Fe and P_4_ vapor according to the Equations (3)–(7). In addition, most of the harmful element fluorine is volatilized in the form of SiF_4_ (g). The yellow P vapor and the ferrophosphorus alloy can be further manufactured to phosphorus series chemical products. Main nutrition, such as 2–3%P, 97% SiO_2_, and 100% CaO, is entered into the molten slag. Fortunately, nutrition in the molten slag can be activated via the subsequent water quenching. The main chemical compositions of the slag are 1–2% P_2_O_5_, 30–45% SiO_2_, 40%−50% CaO, and other micronutrients. After water quenching, the activation efficiencies of P, Ca, and Si are as high as 78.28wt %, 90.03wt %, and 77.12wt %, respectively. In addition, the heavy metal contents of the quenched slag are far less than the fertilizer limitation, which indicates that activated slag made from phosphate ore can be used as an ingredient for rice fertilizer preparation.

Figure 11b shows the possible and recommended industrial equipment for the granular fertilizer preparation from activated YPS. As drawn in Figure 11, the existing reduction smelting equipment is the electric arc furnace in the industrial production of yellow phosphorous. The molten slag is then discharged and flowed into the recirculated cooling water. The activated YPS is further ground in a ball miller to make powdery YPS. The powdery YPS is then mixed with other NPK materials to manufacture serial granular fertilizers based on the plant nutritional requirements via the disc pelletizer. In the feasible industrial application, the commonly used and technically immature ball miller and disc pelletizer are allocated in the yellow phosphorus smelting factory. In general, the foregoing results demonstrate that the granular rice fertilizer prepared from the activated YPS shows superior nutrition activity and mechanical strength. An industrial field test for the fertilizer efficiency of the as-prepared granular rice fertilizer is being conducted in the paddy land, and further results will be reported in the near future.
4Ca_5_(PO_4_)_3_F + 30C + 21SiO_2_ = 3P_4(g)_↑ + 30CO_(g)_↑ + 20CaSiO_3_ + SiF_4(g)_↑ △G ^θ^ = 8938480–5752.18 T (J/mol)(2)C + CO_2(g)_ = 2CO_(g)_, △G^θ^ = 171700–175.82 T (J/mol)(3)3Fe_2_O_3_ + CO_(g)_ = 2Fe_3_O_4_ + CO_2(g)_, △G^θ^ = 32970–53.85 T (J/mol)(4)Fe_3_O_4_ + CO_(g)_ = 3FeO + CO_2(g)_, △G^θ^ = 31500−37.06 T (J/mol)(5)FeO + CO_(g)_ = Fe + CO_2(g)_, △G^θ^ = 13175 + 17.24 T (J/mol)(6)2Fe+P_4__(g)_ = 2FeP_2_, △G^θ^ = −339004 + 146.44 T (J/mol)(7)

## 4. Conclusions

Thermodynamic calculations of SiO_2_-CaO, SiO_2_-CaO-Al_2_O_3_, and SiO_2_-CaO-Al_2_O_3_-MgO systems indicated that ternary acidity m(SiO_2_)/(m(CaO) + m(MgO)) of 0.92 was suitable for the reduction smelting and activation of the slag synchronously. The increase of ternary acidity from 0.75 to 0.92 facilitated the formation of rankinite and cyclowollastonite phases with low melting temperature, which was beneficial to cut down the energy consumption of yellow phosphorus production.The reduction smelting followed by the water quenching process can realize the reduction of phosphorite and activation of YPS synchronously. After smelting at 1450 °C for 60 min with ternary acidity of 0.92, the molten yellow phosphorus slag can be activated by ultrafast water quenching; 78.28% P, 90.03% Ca, and 77.12% Si in the yellow phosphorus slag were activated, which can be readily absorbed by the rice roots. During the water quenching, the molten slag was transformed into microporous, multi-soluble matter and an amorphous structure, which was beneficial to excite the slag to release the micronutrients.Based on the soil and rice planting habits in the Yunnan and Guizhou provinces of China, two kinds of granular rice fertilizers with formulas of NPK (N-P_2_O_5_-K_2_O) 18-8-10 and NPK 20-8-12 were designed. Granular rice fertilizers (2–4 mm) were prepared from the powdery NPK and activated yellow phosphorus slag by a disc granulator. This fertilizer showed higher conversion efficiencies and good mechanical strength. The conversion efficiencies of N, P, and K were over 94%, and the average compressive strengths of the granular fertilizers were over 21N, meeting the requirements of granular rice fertilizer storage and transportation. In addition, the heavy metal contents of the prepared fertilizer were far less than the limitation standard.

## Figures and Tables

**Figure 1 materials-14-02080-f001:**
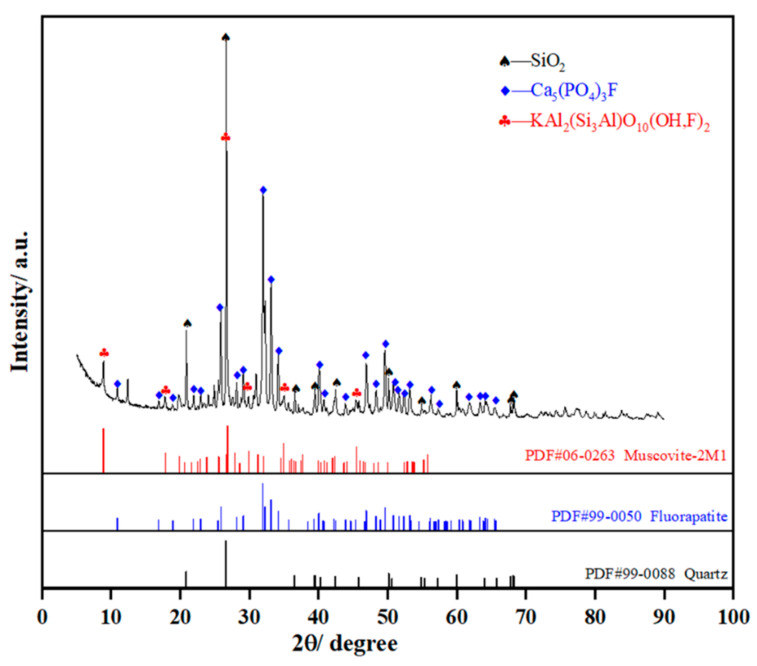
XRD patterns of phosphate ore.

**Figure 2 materials-14-02080-f002:**
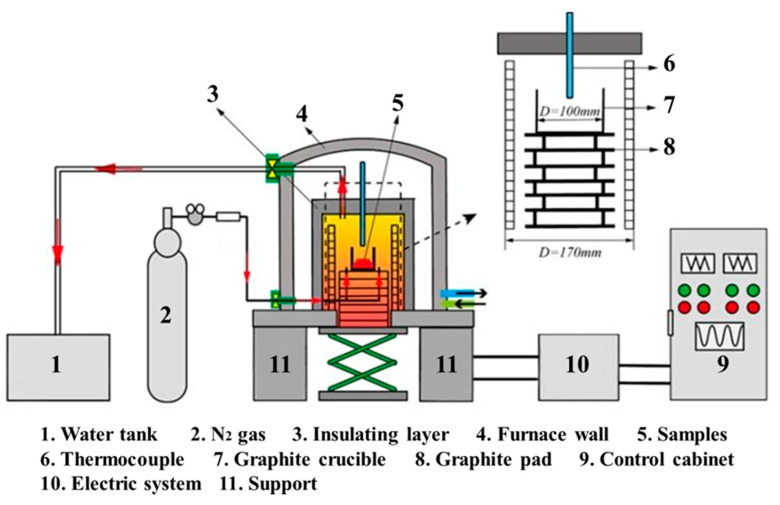
Schematic diagram of the reduction smelting furnace.

**Figure 3 materials-14-02080-f003:**
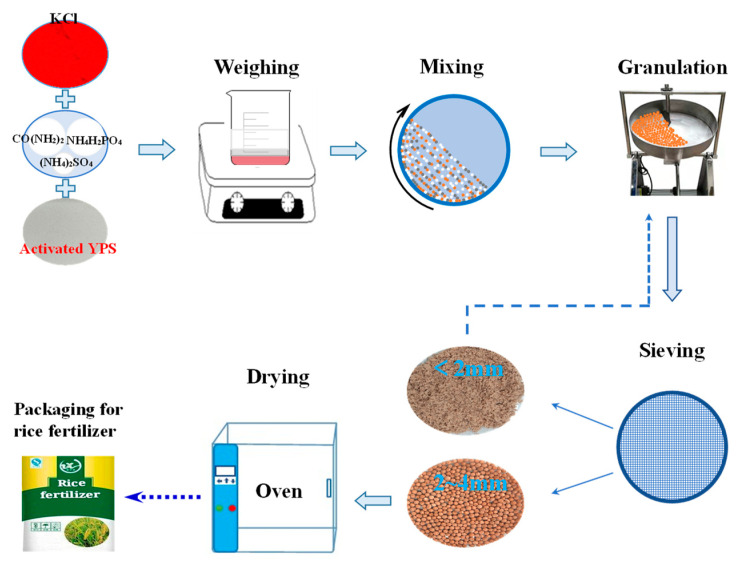
Preparation process of granular rice fertilizer.

**Figure 4 materials-14-02080-f004:**
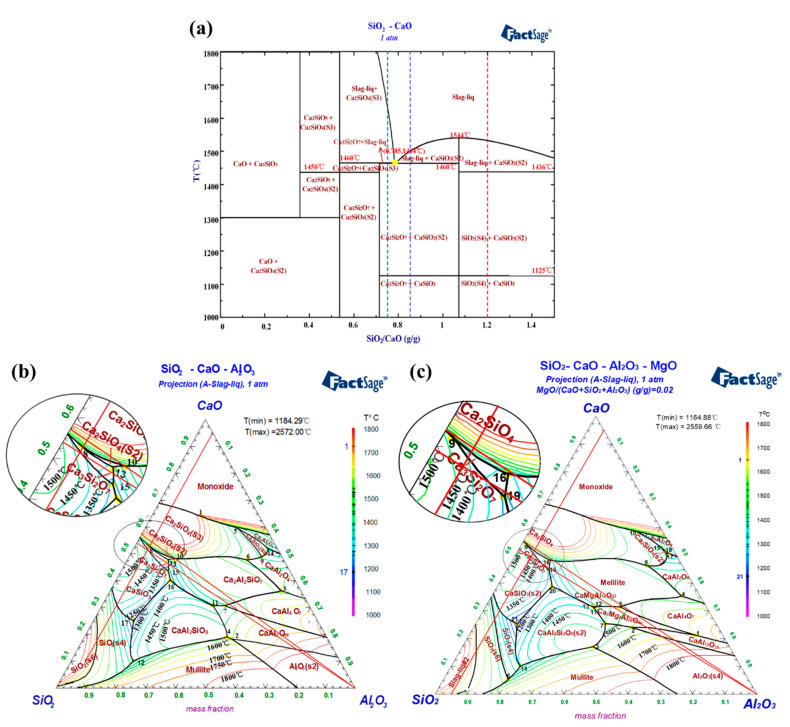
Phase diagrams of SiO_2_-CaO system and SiO_2_-CaO-Al_2_O_3_-MgO system at atmospheric pressure. (**a**) SiO_2_-CaO system; (**b**) SiO_2_-CaO-Al_2_O_3_ system; (**c**) SiO_2_-CaO-Al_2_O_3_-MgO system.

**Figure 5 materials-14-02080-f005:**
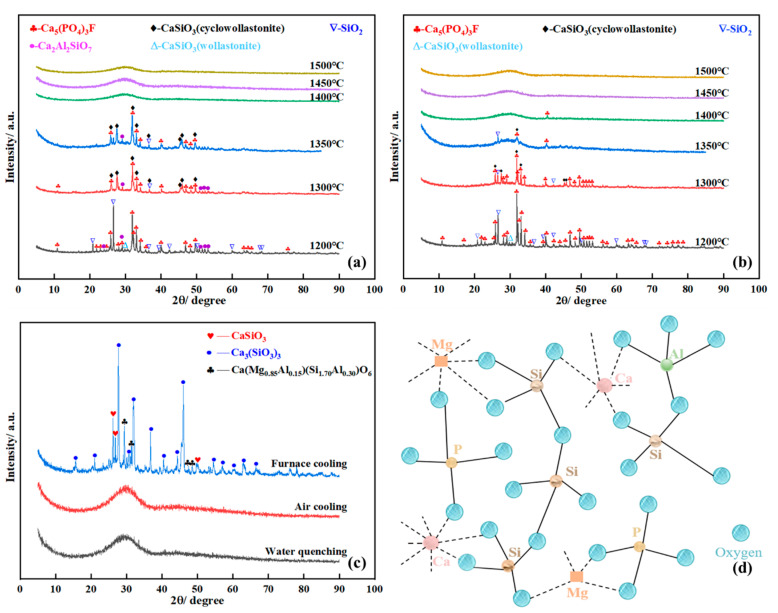
Effect of cooling regimes on the phase transformation of the mixtures with an acidity of 0.92 smelted at different temperatures for 60 min in pure N_2_ (99.99%) atmosphere. (**a**) Air cooling; (**b**) Water quenching; (**c**) Phase comparison of the mixture smelted at 1450 °C for 60 min under different cooling regimes; (**d**) Glass structure of YPS after water quenching.

**Figure 6 materials-14-02080-f006:**
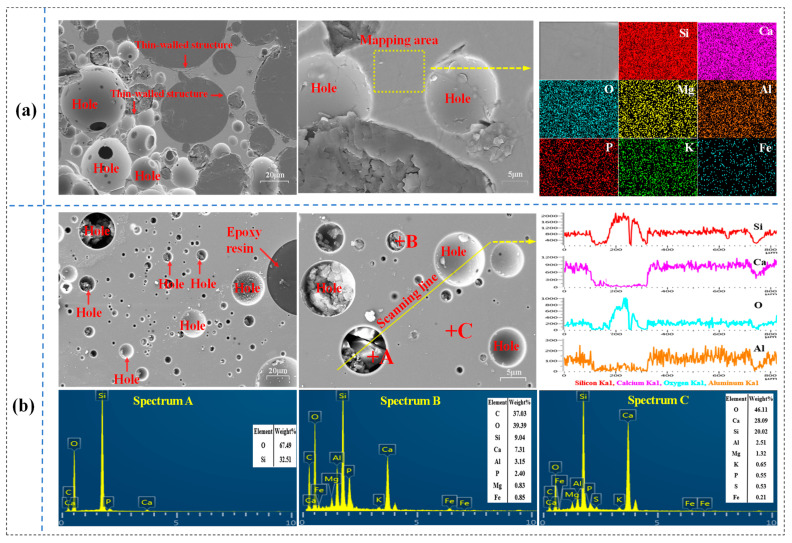
SEM-EDS analyses of the mixtures with an acidity of 0.92 smelted at 1450 °C for 60 min in pure N_2_ (99.99%) atmosphere. (**a**) Water quenching; (**b**) Air cooling.

**Figure 7 materials-14-02080-f007:**
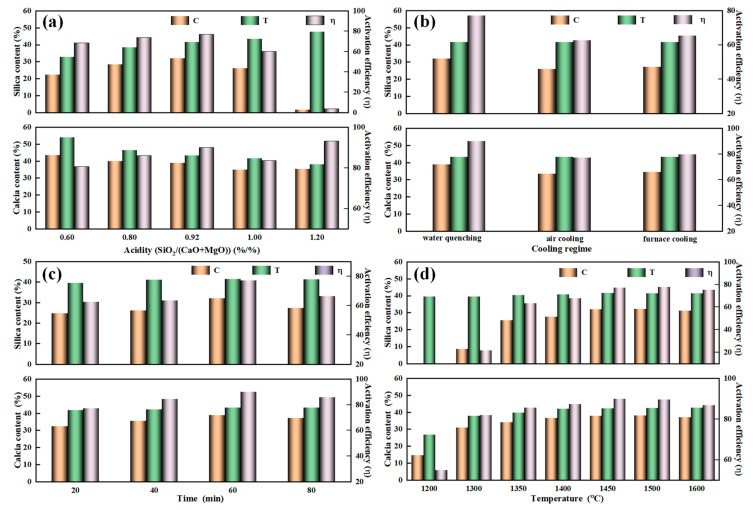
Nutrient activities of the slag obtained under different conditions. (**a**) Smelting at 1450 °C for 60 min under different acidity; (**b**) Smelting at 1450 °C for 60 min with an acidity of 0.92 under different cooling methods; (**c**) Smelting at 1450 °C with an acidity of 0.92 under different reaction time; (**d**) Smelting for 60 min with an acidity of 0.92 under different reaction temperature.

**Figure 8 materials-14-02080-f008:**
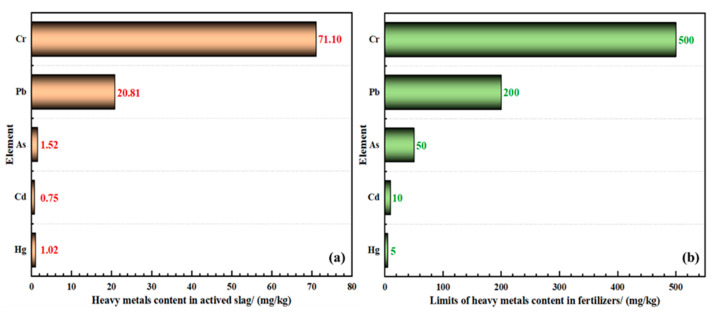
(**a**) Heavy metals leaching toxicity test for active slag; (**b**) Limits of heavy metals content in fertilizers.

**Figure 9 materials-14-02080-f009:**
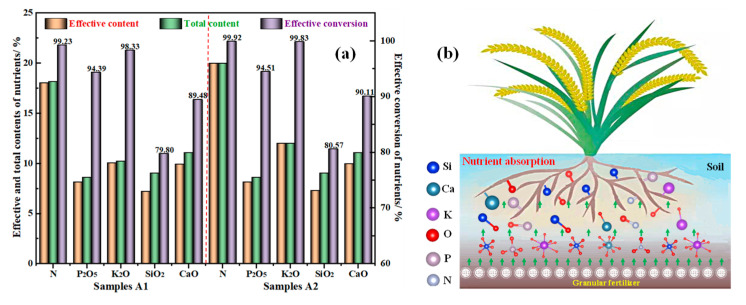
(**a**) Effective conversion of the nutrients in the prepared rice fertilizer (dotted line left is Sample A1, and right is Sample A2); (**b**) Schematic diagram of absorption by the rice roots.

**Figure 10 materials-14-02080-f010:**
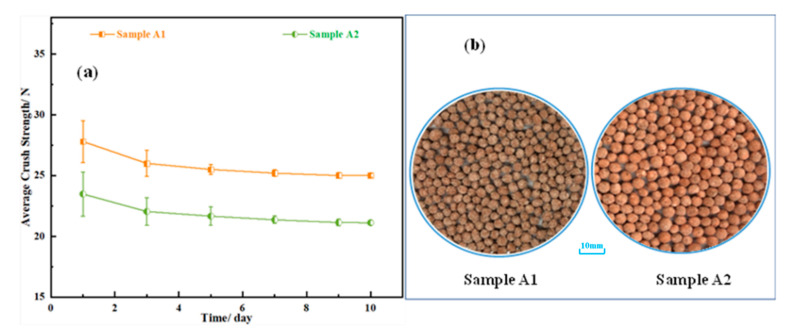
(**a**) Average crush strength of the granular rice fertilizer; (**b**) Appearance of the granular rice fertilizer.

**Figure 11 materials-14-02080-f011:**
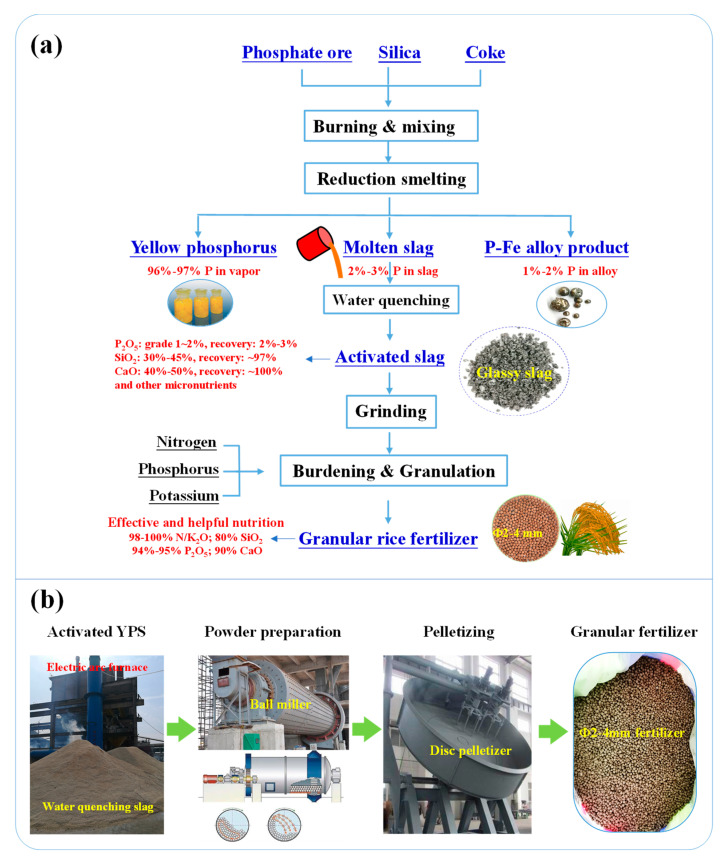
(**a**) Flowsheet and elements distribution in the treatment of phosphorite; (**b**) Possible and recommended industrial equipment for the granular fertilizer preparation from activated YPS in the existing yellow phosphorus smelting factory.

**Table 1 materials-14-02080-t001:** Chemical compositions of the phosphate ore and silica (mass fraction, wt %).

Composition	P_2_O_5_	SiO_2_	CaO	MgO	Fe_2_O_3_	Al_2_O_3_	K_2_O	Na_2_O	MnO
phosphate ore	22.57	29.04	33.72	1.79	1.39	3.63	0.92	0.29	0.10
silica	1.60	87.30	2.22	0.36	0.93	3.35	0.60	0.21	0.01

**Table 2 materials-14-02080-t002:** Chemical compositions of coke ash (mass fraction, wt %).

SiO_2_	CaO	MgO	Fe_2_O_3_	Al_2_O_3_	TiO_2_
37.27	6.09	5.19	20.07	26.64	0.33

**Table 3 materials-14-02080-t003:** Design of raw material compositions under different ternary acidity conditions.

Acidity	Raw Material Compositions, g			
	Phosphate Ore	Silica	Coke	CaO
0.60	37.92	-	6.17	5.91
0.80	42.11	-	6.85	1.04
0.92	41.91	1.20	6.89	-
1.00	40.78	2.52	6.70	-
1.20	38.11	5.63	6.26	-

**Table 4 materials-14-02080-t004:** Rice fertilizer design and formulas.

Sample	Raw Material Compositions, g					
	CO(NH_2_)_2_	(NH_4_)_2_SO_4_	NH_4_H_2_PO_4_	KCl	YPS	Attapulgite
**A1**	310	120	130	170	250	20
**A2**	390	30	130	200	250	0

## Data Availability

The data presented in this study are available on request from the corresponding author.

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
