# Peer review of "Sustainable and Clean Utilization of Yellow Phosphorus Slag (YPS): Activation and Preparation of Granular Rice Fertilizer"

_materials, 2021, doi:10.3390/ma14082080_

Round 1

Reviewer 1 Report

The author’s purpose of the investigation is interesting, also for scientists from related research fields. I would recommend the suggestions described below:

  • The results are not properly described. The authors should first describe in a quantitative manner the data before jump to conclusions. Avoid imprecise terms, such as relatively high or higher melting poiny…but how much? 15% higher? 2-fold?. Avoid jumping immediately to conclusion rather describe property the results
  • The figures are very good but could be globally improved, as possible, once Materials deserves high quality figures and with rigor would avoid lacking of interest for the data. Legends should be also as complete as possible.
  • Thus, I would suggest that in some schemes, numbers could be inserted for a better description of all the steps of the schemes and/or figures. For instance Scheme 2; would not be better clarify to insert numbers on the legends: 1) Water tan; 2) N2 gas cylinder; 3) etc. Please correct the word “insulating” area.
  • Numbers could also be insert in others schemes and/or Figures for clarification: For instance to Fig 3.
  • Please correct the word metals in the Y-axis od figure 8.
  • Fig 10 b, perhaps is missing a scale bar.
  • Figure 11 numbers for each steps of the processes could be insert and referred in the text for clarification.
  • Globally after discussion, the conclusions should followed the order of presentation of the paper with partial conclusions first and then global conclusions.

Reviewer 2 Report

Comments and questions to the authors:

  • The manuscript contains several unsuitable and even misleading expressions of English language. Therefore, the language should be improved and its many phrases not familiar to high-temperature processing or phase equilibria corrected. I recommend you to use a native English speaker with professional engineering background.
  • On p. 2, 2nd paragraph from top, discloses a typical analysis of the YPS waste material and the phosphorus extraction process producing the waste. No attention is, however, paid to the technical details of the state-of-the art of phosphorus extraction, or the phase equilibria dominating the process conditions. This is important background information of the present effort and evidently needed for understanding the outcome of this research. There is a lot of data on phosphorus extraction and removal from steel slags produced in mills using high-P iron ores. An example from the recent literature is, e.g. Journal of Sustainable Metallurgy: https://doi.org/10.1007/s40831-021-00350-6.
  • In Experimental Ch, no information is disclosed about the methods and equipment used in chemical analyses of the materials. The numerical values of the chemical compositions have been reproduced in the manuscript with 4 digits, meaning a huge analytical accuracy of much better than 0.1 % relative. If you aim to present the results with this extremely low uncertainty, the justification including calibrations of the equipment must be given in detail in this chapter.
  • In Ch 2.2. (p. 3), the purity of used nitrogen as protective atmosphere in the reduction furnace must be disclosed.
  • There is nowhere in Ch 2.2 any indication of the inaccuracy and stability of temperature during the reduction experiments, or how temperature was measured as well as how the used temperature probe was calibrated.
  • Tin Table 3, the variable ‘acidity’ is used but the manuscript uses two definitions to the acidity of the slag. Which one is used in here?
  • On p. 5, Ch 2.4, the measurements of the effective contents of the substances in the fertiliser are described. The description is so general that no external scholar is able to repeat your tests. Thus, more detailed explanations are needed.
  • In Ch 2.5, the used external standards in the EDS measurements must be disclosed and the matrix correction procedures done for the primary spectral data.
  • In the Results and Discussion Ch, the big picture of the phase equilibria in the end slag is obtained using the quaternary system CaO-MgO-Al2O3-SiO2 without phosphorus oxide. How representative this approach is? Phosphorus oxides are effectively fluxing the other components of the slag and form strong compounds with some of them. It is therefore not the phosphorus-free matrix only which has an impact on the liquid-solid phase relations at elevated temperatures.
  • Caption of Fig. 4 must contain all the boundary conditions (i.e., the constraints) used in the phase diagram calculations.
  • In Ch 3.2 you use two phrases which need further explanation: what is ‘synchronous reduction’ and how ‘roasting’ carried out at 1450 C differs from ‘melting’ used in the caption of Fig. 7 or from ‘smelting’ described earlier in the Experimental sections?
  • How the glass structure of YPS depicted in Fig. 5d has been determined? That detail must also be disclosed in the Experimental chapters.
  • In Ch 3.3 and later in text you mention a procedure not described earlier in Experimental chapters: ‘the ultrafast cooling’. How was that operation achieved in the present study and how the process can be characterised compared with, e.g., conventional water quenching?
  • 8 is somewhat confusing and gives on a first glimpse an impression of equal concentrations of harmful elements; why you use two ordinate axes with different scales to describe the concentrations?
  • On p. 11 and elsewhere, the expressions ‘18-8-10 (N-P2O5-K2O) and 20-8-12 (N-P2O5-K2O)’ need a suitable reference and possible an explanation in writing.
  • The source(s) of the standard Gibbs energies of reactions (2) to (7) must be disclosed.
  • Where were these Gibbs energy values used in this study? There are no data about the conditions in the smelting crucible disclosed in the manuscript and no operational window of the slag formation process or its outlet gas composition is given to assist the analysis.
  • The manuscript fails to describe the new processing concept described with reference to the phosphorus extraction smelting from where the raw material of these experiments basically were obtained. The question can also be focused on the phosphorus extraction by asking whether it can be modified in such a way that its ‘waste YPS slag’ is similar to this smelting product?

Reviewer 3 Report

This is an excellent paper on the utilization of yellow phosphorus slag as a rice fertilizer. Based on the experimental results, optimal conditions for slag activation in terms of acidity, cooling scheme, reaction time and temperature are proposed.

In this manuscript, the results and discussions are described very clearly. 
In addition to the experimental results, industrial equipment recommended is also illustrated.
It would be better to add a brief explanation on a generation mechanism of CaCO3, shown in Fig.5(c).

Round 2

Reviewer 2 Report

Comment and advice to the authors for the revision:

  • Temperature accuracy pointed out in my detailed comment list as question (5) was not understood correctly in the reply letter. The question is wider than the stability of the used temperature controller along with time where the reply letter focused the technicalities.

In high temperature measurements, the uncertainty of the temperature sensor is typically calibrated by comparing its reading with a known melting point (of IPTS) close to the temperature range of the experiments. Several TC manufacturers provide such calibration data with their products and they typically show an uncertainty of ±2 C in the temperature rage of this study. On top of that, the total uncertainty of the sample temperature includes the inaccuracy of zero-point compensation as well as that of the measuring circuit, including measurements of the reference temperature, which may be up to ±2-3 C. This leads to a total uncertainty of ±4-5 C at 1300-1500 C.

Depending on your calibration procedures, which still are unknown in the manuscript, the total (absolute) temperature uncertainty must be much higher than the reported ±1 C.
